# C-Reactive Protein as a Marker of Inflammation in Children and Adolescents with Metabolic Syndrome: A Systematic Review and Meta-Analysis

**DOI:** 10.3390/biomedicines11112961

**Published:** 2023-11-02

**Authors:** Mihaela-Andreea Podeanu, Adina Turcu-Stiolica, Mihaela Simona Subțirelu, Mioara Desdemona Stepan, Claudiu-Marinel Ionele, Dan-Ionuț Gheonea, Bianca Ștefănița Vintilescu, Raluca Elena Sandu

**Affiliations:** 1Doctoral School, University of Medicine and Pharmacy of Craiova, 200349 Craiova, Romania; podeanu.andreea11@gmail.com; 2Department of Pharmacoeconomics, University of Medicine and Pharmacy of Craiova, 200349 Craiova, Romania; mihaela.subtirelu@yahoo.com; 3Department of Infant Care, Pediatrics and Neonatology, University of Medicine and Pharmacy of Craiova, 200349 Craiova, Romania; desdemona.stepan@umfcv.ro (M.D.S.); vintilescubianca92@gmail.com (B.Ș.V.); 4Department of Gastroenterology, University of Medicine and Pharmacy of Craiova, 200349 Craiova, Romania; ioneleclaudiu@gmail.com (C.-M.I.); dan.gheonea@umfcv.ro (D.-I.G.); 5Department of Biochemistry, University of Medicine and Pharmacy of Craiova, 200349 Craiova, Romania; raluca.sandu@umfcv.ro

**Keywords:** metabolic syndrome, C-reactive protein, inflammation, children, adolescents

## Abstract

Metabolic syndrome (MetS) in the pediatric population has been reported in many studies to be associated with an inflammatory response. However, to our knowledge, there is no definitive conclusion in the form of a meta-analysis. The issue we aimed to address is whether C-reactive protein (CRP) is a trustworthy marker in detecting inflammation in children and adolescents with MetS. We systematically searched PubMed, MEDLINE, Cochrane Central Register of Controlled Trials, the ISI Web of Science, and SCOPUS until 31 June 2023 for studies involving children and adolescents with MetS where hsCRP or CRP were measured. After the screening process, we identified 24 full-text articles that compared 930 patients with MetS with either healthy (n = 3782) or obese (n = 1658) controls. The risk of bias in the included studies was assessed using the Begg’s rank correlation test and Egger’s regression test. Statistical analysis was carried out based on pooled mean differences (MDs) and an associated 95% CI. Data analysis showed that MetS is associated with higher levels of CRP than those in healthy controls (MD = 1.28, 95% CI: (0.49–2.08), *p* = 0.002) in obese patients (MD = 0.88, 95% CI: (0.38–1.39), *p* = 0.0006). However, conventional methods of CRP analysis were found to be more accurate in differentiating between children and adolescents with obesity and those with MetS, compared with hsCRP (MD = 0.60, 95% CI: (−0.08–1.28), *p* = 0.08). No risk of bias was assessed. In conclusion, CRP is a reliable inflammatory marker for differentiating pediatric patients with MetS from healthy ones. On the other hand, it did not prove to be very accurate in distinguishing between patients who had MetS and those who were obese. There should be more research performed in this field.

## 1. Introduction

Metabolic syndrome (MetS) is a risk factor for type 2 diabetes mellitus (T2DM) and cardiovascular diseases (CVDs) [1], and according to some recent data it is strongly associated with nonalcoholic fatty liver disease [2]. Although it has been extensively researched in the adult population, the medical community’s focus has also shifted to the juvenile population in light of the global obesity pandemic that is affecting children and adolescents [3]. Currently, there are no consensus guidelines or diagnostic criteria equivocally accepted for MetS in these populations [4]. However, most of the definitions use the following five entities: obesity, elevated blood pressure, altered blood glucose, elevated triglycerides, and low HDL-cholesterol [5]. The problems encountered are related to the cutoff points and importance of each parameter in the progression of the disease [6,7].

The pathophysiology of MetS is not yet fully understood. It is still unknown if the five elements composing MetS form by themselves distinct pathologies resulting in this destructive process or if it is a combined action [8]. Some authors argue that a high-calorie diet plays a significant role in this pathology, since obesity appears to be a primary factor [9,10,11]. On the other hand, insulin resistance and chronic inflammation are also thought to have leading roles in the progression of MetS and its subsequent transition to CVDs and T2DM [12].

Data gathered from studies performed on adults suggest a degree of chronic low-grade inflammation, which is characterized by cytokine production and activation of inflammatory signaling pathways [13,14]. IL-6 increases the production of acute phase reactants in the liver, including C-reactive protein (CRP). There is a strong correlation between high levels of CRP, CVDs, T2D, and MetS, as shown by multiple studies [12,15,16,17].

CRP is a protein synthetized by the liver. When inflammation is present, its levels rise [18]. It has a long plasma half-life and little to no diurnal variation and it is not age- or gender-dependent, having good stability over time. In healthy individuals it is found in small amounts, but in the presence of infections, autoimmune diseases, or cancer, its levels rapidly increase [19].

Compared with the erythrocyte sedimentation rate and leukocyte count, CRP is highly reliable and it is a more sensitive indicator of acute inflammation [20,21]. There are some conditions, such as obesity, pregnancy, depression, and diabetes, that in some cases are associated with minor elevations in CRP. To detect these low levels, high-sensitivity CRP assay techniques were developed and are recommended to be used as the common techniques are considered to be less accurate [18].

The International Diabetes Federation (IDF) issued in 2007 a guideline on pediatric metabolic syndrome and among other directions in need of research, a call for more investigations regarding the association between high-sensitivity C-reactive protein (hsCRP) and MetS in children was made [22]. In addition, other studies have proposed adding this marker as a clinical criterion for MetS given its additive prognostic value in the prediction of development of T2DM [23]. There are studies that show a significant association between CRP and individual features of MetS and support that adding it to the mandatory investigations will have a beneficial role as it can predict the associated cardiovascular complications [24]. The role of CRP as an early biomarker in predicting progression to T2DM in patients with this pathology is an important point of discussion due to the possibility for early intervention that can slow its progression [25]. Also, the assay for CRP is affordable and generally available in clinical settings [26]. While the correlation between CRP and development of CVDs and T2DM is a well-known fact, the role of this inflammatory marker in the pathogenesis of MetS and its long-term consequences requires further elucidation. Presently, hsCRP is not considered a feature of MetS by any national or global guideline committees [27].

To our knowledge, there have been several original studies published regarding the association between CRP and MetS in children and adolescents, but no conclusion in the form of a meta-analysis has been issued. Our goal is to synthesize the data published on this topic and draw a conclusion regarding the usage of CRP or hsCRP in identifying and monitoring low-grade inflammation in children with MetS.

## 2. Materials and Methods

### 2.1. Search Strategy

In our research, we followed the 27 items of the Systematic reviews and Meta-Analysis (PRISMA) guidelines, and the flow chart was designed according to these guidelines [28]. We registered our work using the INPLASY platform, which is an international platform of registered systematic review and meta-analysis protocols. Our identification number is INPLASY2023100032.

We systematically searched PubMed, SCOPUS, MEDLINE, Cochrane Central Register of Controlled Trials, and the ISI Web of Science for studies on children and adolescents with MetS where hsCRP or CRP was measured.

Two authors (M.A.P. and M.S.S.) independently screened the articles published until 31 May 2023 for inclusion and extracted data. Disagreements were solved by the third author (A.T.S.). The criteria used to select the studies for this meta-analysis were in accordance with the PICO standards (patient, intervention, comparator, outcomes) [29].

We conducted our search using the terms (“Metabolic syndrome” OR “MetS”) AND (“CRP” OR “C reactive protein”) AND (“Children” OR “Adolescents” AND “NOT Adults”).

The information required for our research was extracted by 2 authors (M.A.P. and C.M.I.) using a reliable and standardized method. Disagreements were solved by the third author (A.T.S.). The data collected from each study are introduced in a standardized table. We extracted the following data: DOI, title, authors, year of publication, country, study design, definition used to diagnose MetS, method of CRP or hsCRP analysis, number of participants in total and for each group, and CRP or hsCRP values for each group.

In the cases where CRP or hsCRP was not originally reported in mg/L but in mg/dL, we converted the results in order for all the final results to be expressed in mg/L.

The quality assessment of included studies was performed using the Newcastle–Ottawa quality assessment scale (NOS) for observational studies (cross-sectional, case–control, or cohort), scoring them from 1 to 9. The results were interpreted as follows: low quality (1–3), moderate (4–6), and high quality (7–9) [30,31]. 

### 2.2. Inclusion and Exclusion Criteria

In this meta-analysis, we only included original studies fulfilling the following criteria: (1) clinical study or cohort design; (2) studies on children and/or adolescents with MetS; (3) studies having both a MetS group and a control group with healthy or obese subjects; (4) hsCRP or CRP was measured for both the MetS and the control group; and (5) the MetS group was diagnosed using an internationally accepted definition. For each group of patients, we included only complete study analyses or the most recent largest sample size.

Exclusion criteria included (1) case-report articles, review articles, meta-analyses, abstracts, opinions, or letters; (2) incomplete data or lack of measuring hsCRP or CRP for all the MetS or control subjects; (3) only abstract (without accessible full-text article); (4) studies that examined patients with other chronic or acute pathologies; (5) studies that involved only animals and/or ex vivo samples; (6) studies of low methodological quality; (7) studies with insufficient data; and (8) studies in languages other than English.

In the selection process, there were 3 studies that checked all the inclusion and exclusion criteria but had some missing or not very clear information. In these cases, we contacted the declared corresponding authors, specifying the issue we faced. In one case we received the demanded answer and proceeded to include the study in our analysis. On the other hand, for 2 studies, we did not receive any response, so we decided to exclude them from our analysis as the information was not complete.

### 2.3. Statistical Analysis

Statistical analysis was carried out using pooled mean differences (MDs) and associated 95% CI. The Z test was used for determining the significance of pooled MDs, visually displayed with forest plots. The amount of heterogeneity was estimated using tau^2^, the Q-test, and the I^2^ statistic. In case any amount of heterogeneity was detected (i.e., tau-squared statistic, regardless of the results of the Q-test), a prediction interval for the true outcomes was also provided. A fixed-effects model was considered when heterogeneity was not significant (*p* > 0.05 and I^2^ < 50%). Sensitivity analysis was conducted only if necessary. Tests and confidence intervals were computed using the Knapp and Hartung method. Studentized residuals and Cook’s distances [32] were used to examine whether studies may be outliers and/or influential in the context of the model. Studies with a Cook’s distance larger than the median plus six times the interquartile range of the Cook’s distances are considered to be influential. In addition to calculating publication bias, the Begg’s rank correlation test and Egger’s regression test were applied to check for funnel plot asymmetry (*p*-value < 0.05 indicates significant publication bias). Statistical analyses were performed with Review Manager (Rev Man, version 5.4.1, The Cochrane Collaboration, 2020, Copenhagen) and R package Metafor (version 4.1, R foundation, Vienna, Austria). A *p*-value less than 0.05 was considered statistically significant.

## 3. Results

### 3.1. Study Selection and Characteristics of Included Studies

The association between MetS and CRP in the pediatric population has garnered significant interest among researchers in the last few years. However, to our knowledge, to date, there has been no published study in the form of a meta-analysis on this topic.

To address this problem, we conducted a systematic literature search and identified 430 potential studies. After removing the 91 duplicated studies, we screened the remaining studies using inclusion and exclusion criteria as described in Figure 1. After the screening process, we selected a total of 24 full-text articles, of which 9 measure CRP and 15 hsCRP.

The total number of subjects included in the studies was 6370 children and adolescents aged 6 to 18 years (930 patients with MetS, 1658 patients with obesity, and 3782 healthy controls). In Table 1, we summarize the main information from the included studies, such as the number of participants from each article split according to gender and subgroup. All the studies scored at least seven for the quality assessment (NOS), which proves they were high-quality according to the international accepted ranking. 

Since there is no universally accepted definition for pediatric MetS, we included the studies diagnosing this pathology according to one of the generally accepted definitions [57,58]. The specific criteria and their interpretation as declared by each author are summarized in Table 2.

In order to better understand the issue regarding the inflammatory response in MetS measured through CRP, we considered it necessary to split the studies into four groups according to the characteristics of the control subjects (obese or healthy) and the type of CRP assessed (conventional or hsCRP). The comparison with the control group composed of children and adolescents with obesity was made in order to determine if obesity alone is the cause of these changes or there is a greater inflammatory response in the MetS group.

The results for each category are detailed below.

### 3.2. Meta-Analysis of Studies Measuring hsCRP in MetS Patients Compared with Healthy Controls

The forest plot comparing the hsCRP values among patients with MetS and healthy controls is shown in Figure 2. A total of k = 9 studies were included in the analysis. The observed mean differences ranged from 0.2800 to 9.3000, with the majority of estimates being positive (100%). The estimated average mean difference based on the random-effects model was 2.05 (95% CI: (1.15–2.94)). Therefore, the average outcome differed significantly from zero (Z = 4.47, *p* < 0.00001). According to the Q-test, the true outcomes appear to be heterogeneous (χ2(8) = 378.4428, *p* < 0.0001, tau^2^ = 1.50, I^2^ = 98.0797%). The determination of the 95% prediction interval yielded a range from −1.1798 to 5.3050 in this analysis. Consequently, while the average expected result is positive, there is a possibility that in certain studies, the actual outcome might be negative. A meticulous examination of the standardized residuals indicated that none of the studies exhibited values bigger than ± 2.7729, suggesting no evidence of outliers within the framework of this statistical model. The absence of exceptionally large residuals implies that the data points, as represented by the individual studies, conform reasonably well to the overall trend established by the regression model. This conformity enhances the reliability of the model’s predictions and suggests a consistent pattern across the studies analyzed. Additionally, according to an assessment using Cook’s distances, none of the studies can be regarded as significantly influential. This finding indicates that no single study exerted a disproportionately large impact on the overall regression results. Consequently, the absence of highly influential studies enhances the stability of the model, suggesting that the conclusions drawn from this analysis are less susceptible to the influence of specific data points.

The regression test indicated funnel plot asymmetry (*p* = 0.0255) but the rank correlation test did not (*p* = 0.3585). Based on both funnel plots (Figure 3), but also on the Begg’s test (0.278, *p* = 0.358) and Egger’s test (2.828, *p* = 0.025), there was evidence of publication bias.

### 3.3. Meta-Analysis of Studies Measuring hsCRP in MetS Patients Compared with Obese Patients

A total of k = 12 studies were included in the analysis. The forest plot for comparing their outcomes is shown in Figure 4. The observed standardized mean differences ranged from −1.20 to 6.30, with the majority of estimates being positive (75%). The estimated pooled mean difference based on the random-effects model was 0.60 (95% CI: (−0.08–1.28)). Therefore, the average outcome did not differ significantly from zero (z = 1.73, *p* = 0.08). According to the Q-test, the true outcomes appear to be heterogeneous (χ^2^ (11) = 90.11, *p* < 0.00001, tau^2^ = 1.07, I^2^ = 88%).

The 95% prediction interval for the actual outcomes extends from −1.0421 to 1.7954. Therefore, while the average expected outcome is positive, it is worth noting that in certain studies, the true outcome could indeed be negative. An analysis of the studentized residuals revealed that a single study (Aldhoon-Hainerova 2017_female [46]) exhibited a value exceeding ± 2.8653, suggesting that it might be a potential outlier within the scope of this model. As for Cook’s distances, the analysis indicated that this same study (Aldhoon-Hainerova 2017_female [46]) could be classified as significantly influential. Importantly, neither the rank correlation nor the regression test identified any evidence of asymmetry in the funnel plot (*p* = 0.9466 and *p* = 0.8553, respectively). Based on both funnel plots (Figure 5), but also on the Begg’s test (−0.030, *p* = 0.947) and Egger’s test (−0.182, *p* = 0.855), there was no evidence of publication bias.

### 3.4. Meta-Analysis of Studies Measuring CRP in MetS Patients Compared with Healthy Controls

A total of k = 7 studies were included in the analysis, as shown in Figure 6. The observed standardized mean differences ranged from 0.16 to 2.60, with the majority of estimates being positive (100%). The estimated average standardized mean difference based on the random-effects model was 1.28 (95% CI: (0.49–2.08)). Therefore, the pooled outcome differed significantly from zero (z = 3.16, *p* = 0.002). According to the Q-test, the true outcomes appear to be heterogeneous (Q(6) = 101.16, *p* < 0.00001, tau^2^ = 0.91, I^2^ = 94%).

The determination of the 95% prediction interval for the true outcomes yielded a span from −4.6797 to 9.2448. This interval encapsulates a considerable breadth of potential outcomes, underlining the inherent variability and uncertainty associated with the study’s predictions. While the mean expected outcome leans towards positivity, it is imperative to recognize the possibility of negative outcomes in specific studies. Upon meticulous scrutiny of the studentized residuals, a diagnostic tool employed to identify outliers in regression analysis, a notable observation emerged concerning the study conducted by Zhao in 2019 [49]. The identification of a studentized residual value exceeding ± 2.6901 in this study suggests a departure from the overall trend, marking it as a potential outlier within the confines of this statistical model. Additionally, Cook’s distances, a measure indicating the influence of individual data points on the regression model, revealed that the same study (Zhao 2019 [49]) possessed a significant influence. Crucially, the application of both the Begg’s rank correlation and Egger’s regression tests failed to reveal any evidence of funnel plot asymmetry (*p* = 1.0000 and *p* = 0.6695, respectively), as illustrated in Figure 7.

### 3.5. Meta-Analysis of Studies Measuring CRP in MetS Patients Compared with Obese Patients

The forest plot for the total of k = 5 studies included in the analysis is shown in Figure 8. The observed standardized mean differences ranged from 0.0000 to 2.8, with the majority of estimates being positive (100%). The estimated average standardized mean difference based on the random-effects model was 0.88 (95% CI: (0.38–1.39)). Therefore, the average outcome did not differ significantly from zero (z = 3.41, *p* = 0.0006). According to the Q-test, the true outcomes appear to be heterogeneous (Q(4) = 26.23, *p* < 0.0001, tau^2^ = 0.19, I^2^ = 85%).

The 95% prediction interval for the actual outcomes ranges from −7.5972 to 13.6552. Therefore, even though the average expected outcome is positive, it is essential to acknowledge that in certain studies, the true outcome could actually be negative. Upon inspecting the studentized residuals, it was evident that one study (Kelishadi 2010 [56]) displayed a value exceeding ± 2.5758, suggesting it might be a potential outlier within the framework of this statistical model. Furthermore, the application of Cook’s distances accentuated the significance of the Kelishadi 2010 study [56]. The high value of Cook’s distance associated with this study underscores its potential to disproportionately impact the regression results, underscoring the need for careful consideration and, if necessary, further scrutiny of this particular data point. In addition to these observations, a regression test was conducted to assess the symmetry of the funnel plot. The test yielded a statistically significant result (*p* = 0.0256), indicating the presence of funnel plot asymmetry. Figure 9 visually represents this asymmetry. However, it is noteworthy that the rank correlation test did not yield a significant result (*p* = 0.2333). This discrepancy between the two tests prompts a nuanced interpretation of the findings, suggesting that while there is some evidence of funnel plot asymmetry, further exploration is warranted to discern the precise nature and implications of this asymmetry. In summary, the wide 95% prediction interval underscores the potential variability in study outcomes, necessitating a cautious interpretation of the average expected result. These findings collectively emphasize the complexity of the statistical model under consideration, urging researchers to delve deeper into the data to ensure the reliability and validity of the study’s conclusions.

## 4. Discussion

According to the results we obtained in our meta-analysis, hsCRP and CRP levels were significantly higher in children and adolescents with MetS compared with the healthy controls. This result was consistent with those from other recent studies on both pediatric and adult populations that reported a positive association between this inflammatory biomarker and MetS and consider it a useful predictor of CVDs and T2DM [41,61,62].

Conventional CRP assays are used when evaluating an infection, tissue injury, or inflammatory disorder. Test values are typically considered to be clinically significant at levels above 10 mg/L, which indicate an acute inflammation; on the other hand, in apparently healthy persons and in the general population it should be present in very low levels [21]. High-sensitivity CRP (hs-CRP) assay methods were developed in order to meet the need for a more accurate way to determine the presence of low-grade inflammation [63]. Nephelometric and immunoturbidimetric techniques have been developed to determine these low concentrations [64], but other techniques such as enzyme immunoassays are also being used on a large scale [65].

It is well established that this ability of the hsCRP assay to measure lower ranges from a sample of blood indicates its usefulness in the evaluation of conditions associated with inflammation in otherwise healthy individuals [66], being more sensitive than the conventional CRP assay.

After processing the data we extracted from the included studies, we pointed out that CRP had a better performance in differentiating the MetS group from the simple obesity control group than hsCRP. Even so, these results should be interpreted with caution, as the heterogeneity of the studies included could be the main reason for this outcome.

A recent study comparing the efficacy of conventional CRP and hsCRP measurement techniques found that CRP testing has become more sensitive over time, allowing it to detect lower blood levels [67].

The different definitions used to diagnose MetS in children and adolescents create a flaw in our meta-analysis. There is no international general agreement regarding the definition of this pathology, with recent evidence reporting over 40 definitions recommended by various organizations and authors deriving from criteria used in adults [4,68,69]. Even though the criteria are based on the same five components (obesity, raised triglycerides, low HDL-C, high systolic or diastolic blood pressure, and altered plasma glucose) [70], the slightly different cutoff points for each of them make it difficult to obtain highly relatable data. In such a manner, a unified definition should be taken into discussion as soon as possible.

We observed that in the studies included in our meta-analysis the most used definition was the one stated in 2007 by the International Diabetes Federation [71]. We need to mention that most of the definitions consider obesity as a mandatory criterion for MetS diagnosis [6,22,70].

Many studies over the years have proven the link between obesity and low-grade inflammation [72]. Obesity induces a state of chronic low-grade systemic inflammation in which nutrient overload increases metabolic demands [73]. A study investigating the causal direction of the relationship between excessive adiposity and inflammation, especially focusing on CRP, reached the conclusion that greater adiposity caused by fat mass and obesity-associated genes led to higher CRP levels [74,75]. Adipose tissue is known to produce cytokines that further stimulate the liver to produce CRP, but adipose tissue itself may also be able to produce this protein and, as a result, increase its circulating levels in the blood [76]. The inflammatory profile observed in obese individuals can also be explained by the genetic polymorphisms and interindividual variability in metabolic perturbations associated with excess weight [77].

Nevertheless, we have to take into consideration the fact that hsCRP, according to our findings, was not useful in differentiating children and adolescents with MetS from those with simple obesity. Some of the studies included in our meta-analysis reached the same conclusion [59], but there are also others that found significantly higher levels of hsCRP in those with MetS compared with simple obesity controls matched for age and sex [34]. Bilinski et al. reported that CRP was a good predictor of MetS in males, but not in females. Unfortunately, the designs of the included studies did not allow us to address this issue [50]. Due to the fluctuations in information regarding this topic, and the uncertainty in the data we obtained, we cannot issue a statement regarding the usage of hsCRP in distinguishing obesity from MetS.

## 5. Conclusions

In our meta-analysis, we managed to confirm that C-reactive protein (CRP) is a good marker for identifying inflammation in pediatric metabolic syndrome. Both conventional CRP and highly sensitive CRP analyzing technologies were sensitive in differentiating MetS patients from healthy controls.

An unexpected result of our study was that hsCRP could not differentiate the children and adolescents with MetS from the obesity group. On the other hand, CRP was able to separate these two groups. However, this information is not yet validated as the included studies were heterogenous and more original research studies on larger groups should be conducted in this area.

Taking into consideration the extent of this pathology on a global level, the number of included papers and subjects is not enough for a final conclusion to be drawn.

## Figures and Tables

**Figure 1 biomedicines-11-02961-f001:**
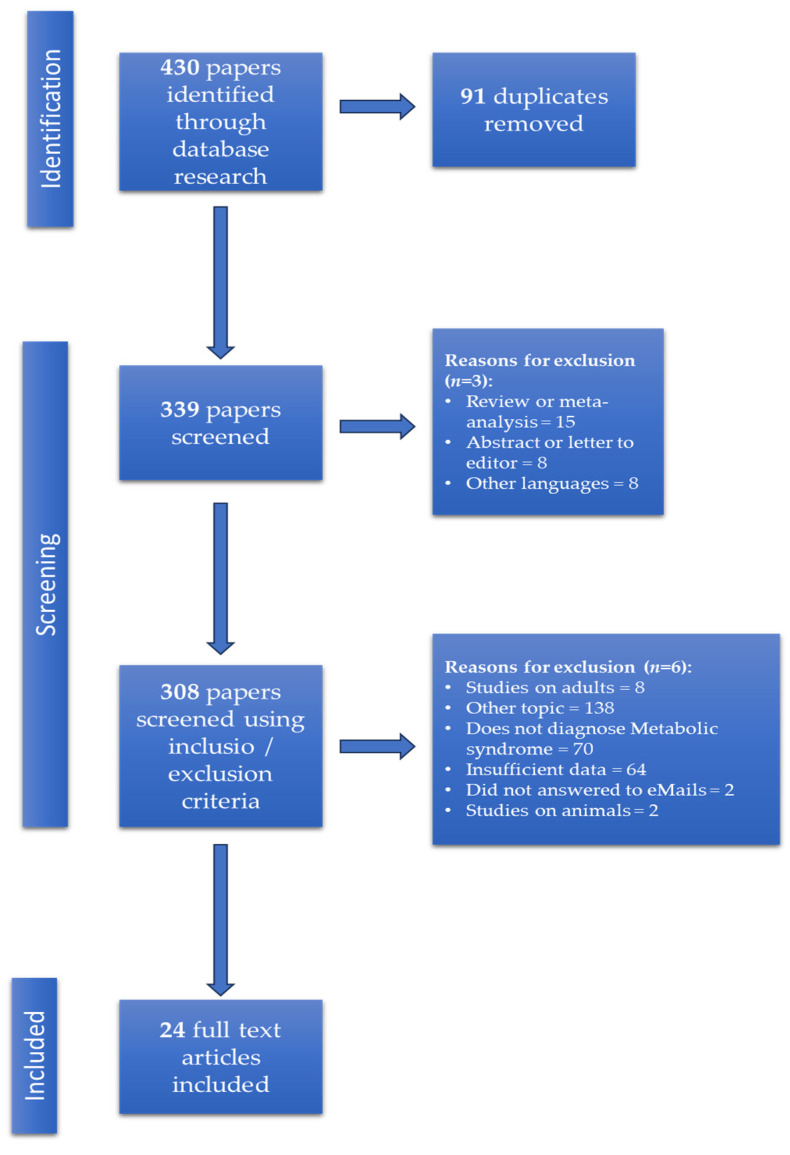
PRISMA flow of the selection process.

**Figure 2 biomedicines-11-02961-f002:**
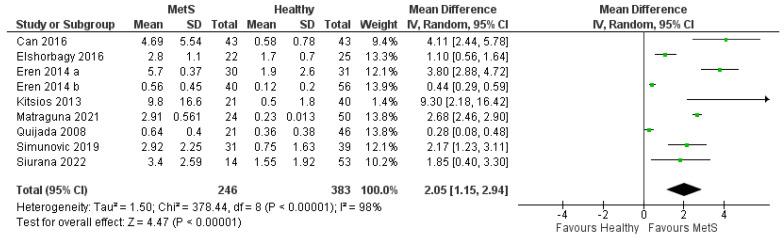
Forest plot assuming a fixed-effects model for hsCRP in MetS vs. healthy controls. The size of the green boxes represents the study weight, the larger the box is, the greater the weight and more information the study provides. The diamond represents the overall pooled effect from the included studies.

**Figure 3 biomedicines-11-02961-f003:**
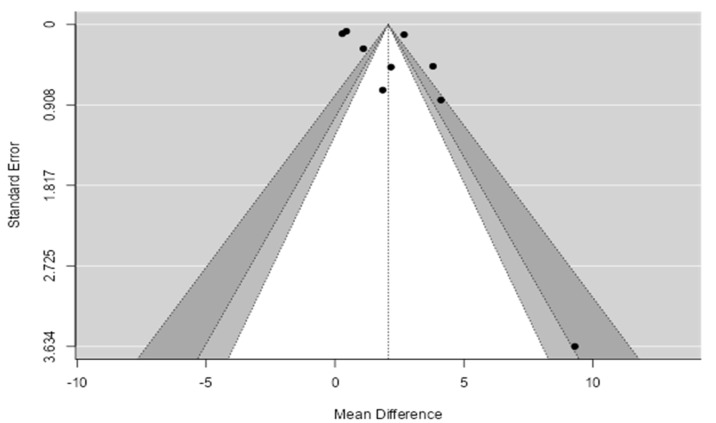
Funnel plot for publication bias assessment for hsCRP in MetS vs. healthy controls. The dots represent the included studies. Various levels of statistical significance of the studies are indicated by the shaded regions: the white region in the middle corresponds to *p* > 0.1, the medium gray-shaded region corresponds to *p* between 0.1 and 0.05, the dark-gray-shaded region corresponds to *p* between 0.05 and 0.01.

**Figure 4 biomedicines-11-02961-f004:**
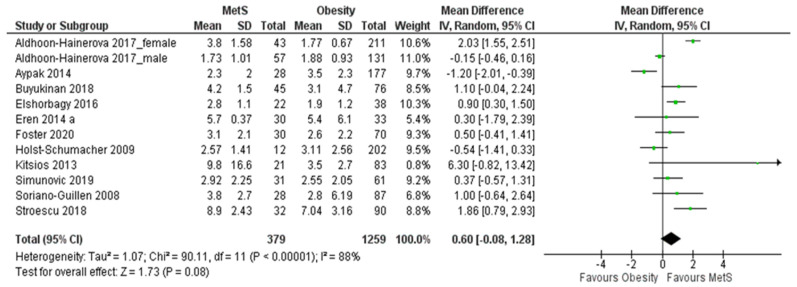
Forest plot assuming a random-effects model for hsCRP in MetS vs. obesity.

**Figure 5 biomedicines-11-02961-f005:**
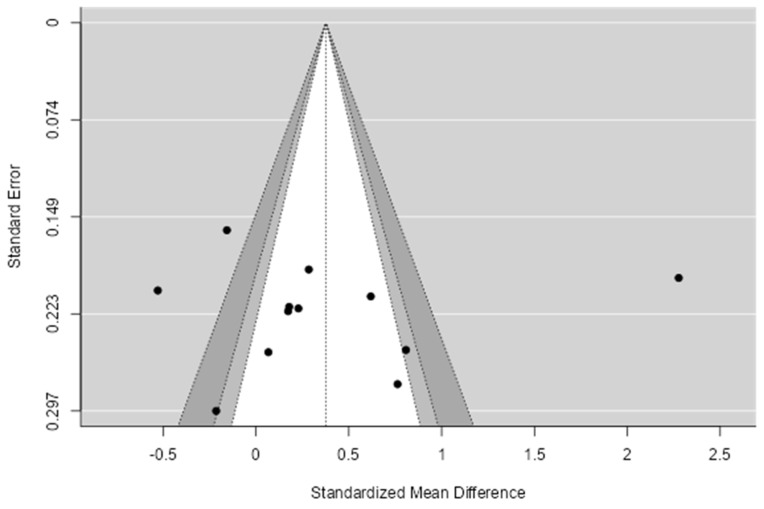
Funnel plot for publication bias assessment for hsCRP in MetS vs. obesity. The dots represent the included studies. Various levels of statistical significance of the studies are indicated by the shaded regions: the white region in the middle corresponds to *p* > 0.1, the medium gray-shaded region corresponds to *p* between 0.1 and 0.05, the dark-gray-shaded region corresponds to *p* between 0.05 and 0.01.

**Figure 6 biomedicines-11-02961-f006:**
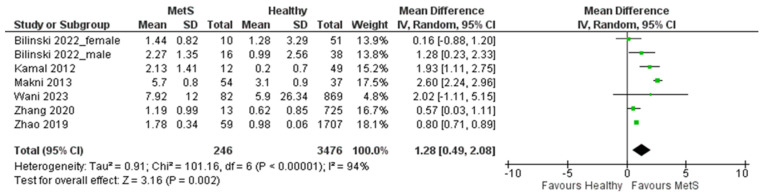
Forest plot assuming a random-effects model for CRP in MetS vs. healthy controls.

**Figure 7 biomedicines-11-02961-f007:**
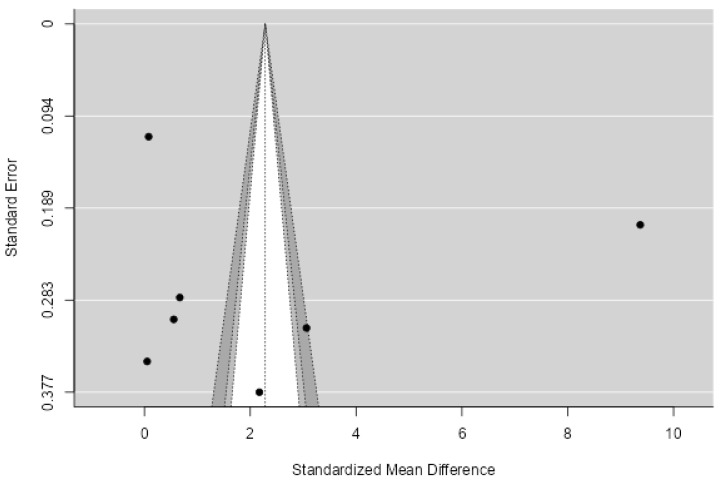
Funnel plot for publication bias assessment for CRP in MetS vs. healthy controls. The dots represent the included studies. Various levels of statistical significance of the studies are indicated by the shaded regions: the white region in the middle corresponds to *p* > 0.1, the medium gray-shaded region corresponds to *p* between 0.1 and 0.05, the dark-gray-shaded region corresponds to *p* between 0.05 and 0.01.

**Figure 8 biomedicines-11-02961-f008:**
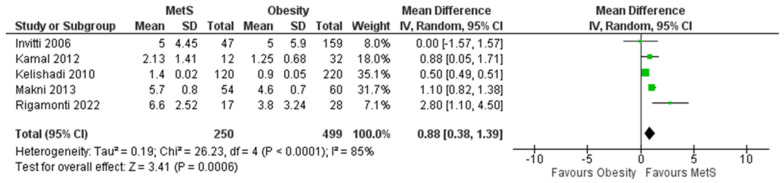
Forest plot assuming a random-effects model for CRP in MetS vs. obesity.

**Figure 9 biomedicines-11-02961-f009:**
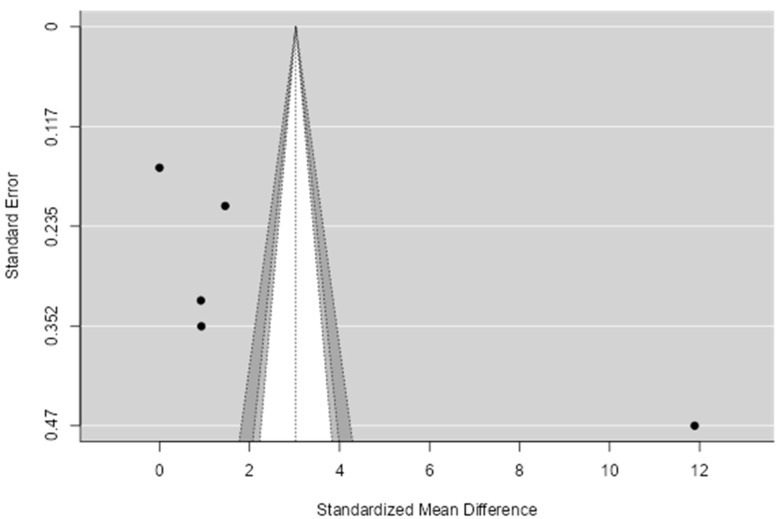
Funnel plot for publication bias assessment for CRP in MetS vs. obesity. The dots represent the included studies. Various levels of statistical significance of the studies are indicated by the shaded regions: the white region in the middle corresponds to *p* > 0.1, the medium gray-shaded region corresponds to *p* between 0.1 and 0.05, the dark-gray-shaded region corresponds to *p* between 0.05 and 0.01.

**Table 1 biomedicines-11-02961-t001:** Characteristics of the studies included in our meta-analysis.

Study	Country	Number of Subjects Included			With MetS	Controls with Obesity	Controls without Obesity	Quality Assessment (NOS)
		Total	Females	Males				
Kitsios et al., 2013 [33]	Greece	144	64	80	21	83	40	8
Matraguna et al., 2021 [34]	Republic of Moldova	74	-	-	24	-	50	7
Eren et al., 2014 a * [35]	Turkey	94	44	50	30	31	33	8
Siurana et al., 2022 [36]	Spain	67	-	-	14	-	53	7
Eren et al., 2014 b * [37]	Turkey	96	40	56	40	-	56	8
Can et al., 2016 [38]	Turkey	86	38	58	43	-	43	8
Simunovic et al., 2019 [39]	Croatia	131	70	61	31	61	39	8
Elshorbagy et al., 2016 [40]	Egypt	60	-	-	22	38	-	8
Stroescu et al., 2018 [41]	Romania	122	49	73	32	92	-	8
Buyukinan et al., 2018 [42]	Turkey	121	42	79	45	76	-	8
Soriano-Guillen et al., 2008 [43]	Spain	115	-	-	28	87	-	8
Holst-Schumacher et al., 2009 [44]	Costa Rica	214	110	104	12	202	-	8
Aypak et al., 2014 [45]	Turkey	205	-	-	28	177	-	8
Aldhoon-Hainerová et al., 2017 [46]	Czech Republic	442	188	254	100	342	-	7
Foster et al., 2020 [47]	United States of America	100	43	57	30	70	-	8
Kamal et al., 2012 [48]	Egypt	93	53	40	12	32	49	8
Zhao et al., 2019 [49]	China	1766	871	895	59	-	1707	8
Bilinski et al., 2022 [50]	Poland	115	54	61	26	-	81	9
Makni et al., 2013 [51]	Tunisia	151	76	75	54	60	37	9
Wani et al., 2023 [52]	Saudi Arabia	951	503	448	82	-	869	8
Zhang et al., 2020 [53]	China	738	-	-	13	-	725	9
Invitti et al., 2006 [54]	Italy	206	-	-	47	159		8
Rigamonti et al., 2022 [55]	Italy	45	17	28	17	28	-	8
Kelishadi et al., 2009 [56]	Iran	240	-	-	120	120	-	7

* Eren E. (Eren et al. [35,37]) was the first author for 2 studies approaching 2 different topics in the same year. In order to distinguish them, we marked them a and b.

**Table 2 biomedicines-11-02961-t002:** Definitions and their modifications used to diagnose metabolic syndrome in the included studies.

Study	Criteria Used to Diagnose Metabolic Syndrome
Kitsios et al., 2013 [33]	Modified Cook criteria: • Fasting glucose levels > 100 mg/dL • Waist circumference values were plotted based on the centiles established by Fernandez et al. [59] for US children and adolescents of European origin, since there are no published reference data for the Greek population • Elevated systolic and diastolic blood pressure ≥ 90th percentile for age, sex, and height or previously diagnosed hypertension • Triglycerides ≥ 110 mg/dL (≥1.24 mmol/L) • HDL-cholesterol ≤ 40 mg/dL (≤1.03 mmol/L)
Matraguna et al., 2021 [34]	IDF criteria (2007): • Central obesity (WC): ≥90th percentile or adult cutoff if lower and at least two of the following criteria • Triglycerides ≥ 1.7 mmol/L (≥150 mg/dL) • HDL-cholesterol < 1.03 mmol/L (<40 mg/dL) • Blood pressure: systolic BP ≥ 130 or diastolic BP ≥ 85 mmHg (or treatment of previously diagnosed hypertension) • Fasting plasma glucose ≥ 5.6 mmol/L (100 mg/dL) (or previously diagnosed type 2 diabetes)
Eren et al., 2014 a [35]	IDF criteria: • Central obesity (WC) and at least two of the following criteria • Triglycerides ≥ 1.7 mmol/L (150 mg/dL) • HDL-cholesterol < 1.03 mmol/L (40 mg/dL) in males and <1.29 mmol/L (50 mg/dL) in females (or specific treatment for these lipid abnormalities) • Blood pressure: systolic BP ≥ 130 or diastolic BP ≥ 85 mmHg (or treatment of previously diagnosed hypertension) • Fasting plasma glucose ≥ 5.6 mmol/L (100 mg/dL) (or previously diagnosed type 2 diabetes)
Siurana et al., 2022 [36]	Cook et al. [60]: • Waist circumference ≥ 90th percentile for age and sex • Elevated systolic and diastolic blood pressure ≥ 90th percentile for age, sex, and height or previously diagnosed hypertension • Fasting glucose levels ≥ 110 mg/dL (≥6.1 mmol/L) • Triglycerides ≥ 110 mg/dL (≥1.24 mmol/L) • HDL-cholesterol ≤ 40 mg/dL (≤1.03 mmol/L)
Eren et al., 2014 b [37]	IDF criteria: • Central obesity (WC) and at least two of the following criteria • Triglycerides ≥ 1.7 mmol/L (≥150 mg/dL) • HDL-cholesterol < 1.03 mmol/L (<40 mg/dL) and <1.29 mmol/L (50 mg/dL) in females (or specific treatment for these lipid abnormalities) • Blood pressure: systolic BP ≥ 130 or diastolic BP ≥ 85 mmHg (or treatment of previously diagnosed hypertension) • Fasting plasma glucose ≥ 5.6 mmol/L (100 mg/dL) (or previously diagnosed type 2 diabetes)
Can et al., 2016 [38]	IDF criteria: • Central obesity (WC) and at least two of the following criteria • Triglycerides ≥ 1.7 mmol/L (≥150 mg/dL) • HDL-cholesterol < 1.03 mmol/L (<40 mg/dL) and <1.29 mmol/L (50 mg/dL) in females (or specific treatment for these lipid abnormalities) • Blood pressure: systolic BP ≥ 130 or diastolic BP ≥ 85 mmHg (or treatment of previously diagnosed hypertension) Fasting plasma glucose ≥ 5.6 mmol/L (100 mg/dL) (or previously diagnosed type 2 diabetes)
Simunovic et al., 2019 [39]	IDF criteria (2007): • Central obesity (WC): ≥90th percentile or adult cutoff if lower (from 10 to 16 years old) • WC >80 cm for women and >94 cm for men (>16 years old) and at least two of the following criteria • Triglycerides ≥ 1.7 mmol/L (≥150 mg/dL) • HDL-cholesterol <1.03 mmol/L (<40 mg/dL) • Blood pressure: systolic BP ≥ 130 or diastolic BP ≥ 85 mmHg (or treatment of previously diagnosed hypertension) • Fasting plasma glucose ≥ 5.6 mmol/L (100 mg/dL) (or previously diagnosed type 2 diabetes)
Elshorbagy et al., 2016 [40]	IDF criteria: • Central obesity and at least two of the following criteria • Triglycerides ≥ 1.7 mmol/L (≥150 mg/dL) • HDL-cholesterol < 1.03 mmol/L (<40 mg/dL) and <1.29 mmol/L (50 mg/dL) in females (or specific treatment for these lipid abnormalities) • Blood pressure: systolic BP ≥ 130 or diastolic BP ≥ 85 mmHg (or treatment of previously diagnosed hypertension) Fasting plasma glucose ≥ 5.6 mmol/L (100 mg/dL) (or previously diagnosed type 2 diabetes)
Stroescu et al., 2018 [41]	Weiss et al.: • Obesity and at least two of the following criteria • Triglycerides above the 95th percentile • HDLc under the 5th percentile adjusted for age and sex • Elevated systolic and diastolic blood pressure values that exceed the 95th percentile for age and sex • Glycemia (oral glucose tolerance test (OGTT)) of 140–200 mg/dL
Buyukinan et al., 2018 [42]	IDF criteria: • Central obesity: WC ≥ 90th percentile or adult cutoff if lower and at least two of the following criteria • Triglycerides ≥ 1.7 mmol/L (150 mg/dL) • HDL-cholesterol < 1.03 mmol/L (40 mg/dL) in males and <1.29 mmol/L (50 mg/dL) in females (or specific treatment for these lipid abnormalities) • Blood pressure: systolic BP ≥ 130 or diastolic BP ≥ 85 mmHg (or treatment of previously diagnosed hypertension) Fasting plasma glucose ≥ 5.6 mmol/L (100 mg/dL) (or previously diagnosed type 2 diabetes)
Soriano-Guillen et al., 2008 [43]	• Obesity: BMI > 2 SDS for age and sex according to Spanish BMI data and at least two of the following criteria • HDL-cholesterol < 5th percentile • Triglycerides > 95th percentile for age and sex • Diastolic and/or systolic blood pressure higher than 95th percentile for age, sex, and height • Alteration in glucose metabolism according to criteria of the American Society of Diabetes (fasting plasma glucose ≥ 5.6 mmol/L (100 mg/dL))
Holst-Schumacher et al., 2009 [44]	Tapia-Ceballos criteria: • Triglycerides ≥ 110 mg/dL (≥1.24 mmol/L) • HDL-cholesterol < 40mg/dL (<1.03 mmol/L) • Fasting glucose (≥5.55 mmol/L) • Waist circumference ≥ 90th percentile for age and sex • Elevated blood pressure ≥ 90th percentile for age, sex, and height
Aypak et al., 2014 [45]	National Cholesterol Education Program Adult Treatment Panel III: • Abdominal obesity (waist circumference): >102 cm in men and >88 cm in women • Triglycerides ≥ 150 mg/dL • HDL-cholesterol: <40 mg/dL in men and <50 mg/dL in women • Blood pressure: ≥130/≥85 mmHg • Fasting plasma glucose ≥ 110 mg/dL
Aldhoon-Hainerová et al., 2017 [46]	IDF criteria:• Obesity (BMI > 97 percentile; waist circumference 10–16 years: ≥ 90.0 percentile or adult 25 cutoff if lower; >16 years: ≥ 94 cm for boys and ≥ 80 cm for girls) and at least two of the following criteria • Triglycerides ≥ 1.7 mmol/L (≥150 mg/dL) • HDL-cholesterol < 1.03 mmol/L (<40 mg/dL) for individuals 13–15.9 years and boys ≥ 16 years and <1.29 mmol/L (50 mg/dL) in girls ≥16 years • Blood pressure: systolic BP ≥ 130 or diastolic BP ≥ 85 mmHg (or treatment of previously diagnosed hypertension) • Fasting plasma glucose ≥ 5.6 mmol/L (100 mg/dL) (or previously diagnosed type 2 diabetes)
Foster et al., 2020 [47]	IDF criteria: • Central obesity (defined as a waist circumference > 95th percentile) and at least two of the following criteria • Triglycerides ≥ 150 mg/dL • HDL-C < 40 mg/dL in males or < 50 mg/dL in females • blood pressure: BP > 95th percentile based on height, age, and gender • Fasting plasma glucose >100 mg/dL
Kamal et al., 2012 [48]	National Cholesterol Education Program Adult Treatment Panel III: • BMI >85th percentile • Triglycerides ≥ 110 mg/dL • HDL-cholesterol: < 40 mg/dL • Systolic or diastolic blood pressure (>90th percentile) • Fasting plasma glucose ≥ 110 mg/dL
Zhao et al., 2019 [49]	Central obesity + 2 other conditions: • Central obesity as measured using the WHtR was adopted in this study (≥ 0.46 for girls and ≥ 0.48 for boys) • Triglycerides ≥ 110 mg/dL (>1.47 mmol/L) • HDL-cholesterol: < 40 mg/dL (1.03 mmol/L) • Systolic blood pressure ≥ 130 mmHg or diastolic blood pressure ≥ 85 mmHg • Fasting plasma glucose ≥ 100 mg/dL (5.6 mmol/L)
Bilinski et al., 2022 [50]	National Cholesterol Education Program Adult Treatment Panel III: • Waist circumference ≥ 90th percentile of WC by sex and age for European population • Triglycerides ≥ 110 mg/dL • HDL-cholesterol: <40 mg/dL • Systolic or diastolic blood pressure (>90th percentile) • Fasting plasma glucose ≥ 100 mg/dL
Makni et al., 2013 [51]	IDF criteria: • Waist circumference ≥ 90th percentile and at least two of the following criteria • Triglycerides ≥ 1.7 mmol/L (≥150 mg/dL) • HDL-cholesterol <1.03 mmol/L (<40 mg/dL) • Blood pressure: systolic BP ≥ 130 or diastolic BP ≥ 85 mmHg • Fasting plasma glucose ≥ 5.6 mmol/L (100 mg/dL)
Wani et al., 2023 [52]	Cook et. al.: • Elevated waist circumference: age-specific waist circumference of ≥90th percentile • Elevated blood pressure: age-specific systolic or diastolic blood pressure of ≥90th percentile • Elevated fasting glucose: fasting glucose level of ≥6.1 mmol/L • Elevated triglycerides: circulating triglyceride levels of ≥1.24 mmol/L for age 10–15 years and ≥1.7 mmol/L for age ≥16 years • Low HDL-cholesterol: circulating HDL-cholesterol level of ≤1.03 mmol/L
Zhang et al., 2020 [53]	IDF and AAP modified criteria (3 or more): • Obesity: waist ≥ 95th percentile of children of the same age and gender, or BMI ≥ 95th percentile of children of the same age and gender • Dyslipidemia: (a) reduced HDL-C (<1.03 mmol/L) or (b) elevated TG (≥1.47 mmol/L) • Hypertension: blood pressure ≥95th percentile of children of the same age and gender (fast identified: systolic BP ≥120 mmHg or diastolic BP ≥80 mmHg) • Hyperglycemia: fasting glucose ≥ 5.6 mmol/L
Invitti et al., 2006 [54]	WHO adult definition with modifications for children: • Glucose intolerance and 2 or more criteria • Triglycerides >95th percentile of controls • HDL-cholesterol < 5th percentile • Systolic or diastolic blood pressure > 95th percentile • Waist circumference or BMI >97th percentile of controls
Rigamonti et al., 2022 [55]	IDF criteria: • Waist circumference ≥ 90th percentile for ages <16 years and ≥94 cm for males and ≥80 cm for female for ages >16 years and at least two of the following criteria • Triglycerides ≥ 1.7 mmol/L (≥150 mg/dL) for ages < 16 years and the same cutoff or specific treatment for this lipid abnormality for ages > 16 years • HDL-cholesterol < 1.03 mmol/L (<40 mg/dL) for males and females for ages < 16 years and <40 mg/dL for males and <50 mg/dL (1.29 mmol/L) for females or specific treatment for this lipid abnormality for ages > 16 years • Blood pressure: systolic BP ≥ 130 or diastolic BP ≥ 85 mmHg for ages < 16 years and the same cutoff or treatment of previously diagnosed hypertension for ages > 16 years • Fasting plasma glucose ≥ 5.6 mmol/L (100 mg/dL) or previously diagnosed type 2 diabetes mellitus for all ages
Kelishadi et al., 2010 [56]	IDF criteria: • waist circumference ≥ 90th percentile and at least two of the following criteria • Triglycerides ≥ 1.7 mmol/L (≥150 mg/dL) • HDL-cholesterol < 1.03 mmol/L (<40 mg/dL) • Blood pressure: systolic BP ≥ 130 or diastolic BP ≥ 85 mmHg • Fasting plasma glucose ≥ 5.6 mmol/L (100 mg/dL)

## Data Availability

Not applicable.

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
