# Peer review of "C-Reactive Protein as a Marker of Inflammation in Children and Adolescents with Metabolic Syndrome: A Systematic Review and Meta-Analysis"

_biomedicines, 2023, doi:10.3390/biomedicines11112961_

Round 1
Reviewer 1 Report
Comments and Suggestions for Authors
1. The conclusion that CRP is a superior indicator for the differentiation of metabolic syndrome from obese controls in comparison with hsCRP should be interpreted with caution. As described in the limitations, metabolic syndrome and obesity have significant similarities with only differences in diagnostic methods. The prevalence of co-occurrence varies in the literature, but is usually higher than 50%, and if this study did not distinguish between the two, the conclusion that CRP is a superior marker would be unwarranted.
2. If possible, I would like to know the authors' opinion on comparing hsCRP or CRP by dividing subgroups in this study by gender or race, IDF definition differences, etc.
3. I was wondering why the authors removed abstract from meta-analysis.
Thank you.
Comments on the Quality of English Language
English editting is required.
Author Response
We extend our sincere gratitude for the constructive comments provided by you. Your insightful feedback has been invaluable in refining our manuscript. We also appreciate the time and effort you dedicated to reviewing our work thoroughly.
In response to your comments, we have diligently addressed each point raised, implementing necessary corrections in our manuscript. These revisions have been meticulously made using the tracked changes feature, ensuring transparency and traceability in the editing process. Your feedback has significantly contributed to the improvement of the quality and clarity of our research, and we are grateful for your input.
Question 1: The conclusion that CRP is a superior indicator for the differentiation of metabolic syndrome from obese controls in comparison with hsCRP should be interpreted with caution. As described in the limitations, metabolic syndrome and obesity have significant similarities with only differences in diagnostic methods. The prevalence of co-occurrence varies in the literature, but is usually higher than 50%, and if this study did not distinguish between the two, the conclusion that CRP is a superior marker would be unwarranted.
Answer 1: Thank you for your observation. After carefully reading the paragraphs where we addressed the issue regarding the performance of CRP in differentiating children and adolescents with obesity from the ones with Metabolic syndrome, we noticed the formulation was not very clear. We remodeled those paragraphs in order to be more accurate as follows:
Abstract: “In conclusion, CRP is a reliable inflammatory marker for differentiating pediatric patients with MetS from healthy ones. On the other hand, it did not prove to be very accurate in distinguishing between patients who had MetS and those who were obese. There should be more research done in this field.”
Discussion: “After processing the data we extracted from the included studies, we pointed out that CRP had a better performance in differentiating the MetS group from the simple obesity control group than hsCRP. Even so, these results should be interpreted with caution, as the heterogeneity of the studies included could be the main reason for this outcome.
According to a recent study comparing the effectiveness of traditional CRP and hsCRP measurement methods, CRP testing has improved over time, and it is now sensitive enough to identify lower blood levels.”
Conclusion: “An unexpected result of our study was that hsCRP could not differentiate the children and adolescents with MetS from the obesity group. On the other hand, CRP was able to separate these two groups. However, this information is not yet to be validated as the included studies were heterogenous and more original research studies on larger groups should be conducted in this area.”
Question 2: If possible, I would like to know the authors' opinion on comparing hsCRP or CRP by dividing subgroups in this study by gender or race, IDF definition differences, etc.
Answer 2: Your suggestion is extremely interesting, and it would be a great addition to our work. When designing the study, we also took into consideration to analyze the groups by gender, unfortunately, during the process of data extraction we run into the conclusion that it would not lead to accurate results as most of the authors did not report the data in this manner (for example, in a gender subgroup analysis, we need the mean±SD hsCRP of male MetS, respectively Obesity patients, and female MetS, respectively Obesity patients).
However, there was an article in which the subjects were divided into groups on the basis of gender. The results of this study showed that for male subjects CRP levels were a good predictor of MetS, but no such association was found for female participants.
The study:
Bilinski, W.J.; Stefanska, A.; Szternel, L.; Bergmann, K.; Siodmiak, J.; Krintus, M.; Paradowski, P.T.; Sypniewska, G. Relationships between Bone Turnover Markers and Factors Associated with Metabolic Syndrome in Prepubertal Girls and Boys. Nutrients 2022, 14, 1205, doi:10.3390/nu14061205.
After careful consideration, we decided to add in the discussion section the before mentioned conclusion of the study as it is the closest change, we could make in addressing this issue. The added information is: “Bilinski et al. reported that CRP was a good predictor of MetS in males, but not in females. Unfortunately, the designs of the included studies did not allow us to address this issue.”
Your suggestion of dividing the included studies taking into consideration the protocol used to diagnose MetS is extremely interesting. We did not tackle this idea in our manuscript because of the small number of studies that will fall in each category (hsCRP vs Healthy, hsCRP vs Obesity, CRP vs Healthy, CRP vs Obesity).
Question 3: I was wondering why the authors removed abstract from meta-analysis.
Answer 3: We believe there was some misunderstanding. In the version we sent to be revised, the abstract was attached. Below, you can find the abstract of our meta-analysis. We must mention that the conclusion was altered from the original version, at the suggestion of Reviewer 2, who noticed a flaw in this section.
“Abstract: Metabolic syndrome (MetS) in the pediatric population has been reported in many studies to be associated with an inflammatory response. However, to our knowledge, there is no definitive conclusion in the form of a meta-analysis. The issue we aimed to address is whether C-reactive protein (CRP) is a trustworthy marker in detecting inflammation in children and adolescents with MetS. We systematically searched PubMed, MEDLINE, Cochrane Central Register of Controlled Trials, the ISI Web of Science, and SCOPUS until 31 June, 2023, for studies involving children and adolescents with MetS, where hsCRP or CRP were measured. After the screening process, we identified 24 full text articles that compared 930 patients with MetS with either healthy (n=3782) or obese (n=1658) controls. The risk of bias in the included studies was assessed using the rank correlation test Begg’s and Egger’s Regression tests. Statistical analysis was carried out by pooled mean differences (MD) and associated 95% CI. Data analysis showed that MetS is associated with higher levels of CRP than Healthy (MD=1.28, 95% CI: (0.49 - 2.08), p = 0.002) of obese patients (MD=0.88, 95% CI: (0.38 - 1.39), p = 0.0006). However, conventional methods of CRP analysis were found to be more accurate in differentiating between children and adolescents with obesity and those with MetS, compared to hsCRP (MD=0.60, 95% CI: (-0.08 - 1.28), p = 0.08). No risk of bias was assessed. In conclusion, CRP is a reliable inflammatory marker for differentiating pediatric patients with MetS from healthy ones. On the other hand, it did not prove to be very accurate in distinguishing between patients who had MetS and those who were obese. There should be more research done in this field.
Reviewer 2 Report
Comments and Suggestions for Authors
Dear Authors,
Congratulation for you article, we appreciated the clarity, the pertinence and your honesty in the conclusions. We have only few observations.
Line 75 hsCRP, please put before High Sensitivity C-Reactive Protein
Line 99 please take out until 31 May 2023 to avoid repetition
We wish you success in the publication
Author Response
The authors express their sincere gratitude to the reviewer for their thoughtful feedback, kind words, and valuable suggestions, all of which significantly contributed to enhancing the quality of our work. The meticulous revisions and refinements recommended by the reviewer were incorporated into the paper using the "Track Changes" feature.
Reviewer 3 Report
Comments and Suggestions for Authors
The authors performed a systematic review and meta-analysis of published research to address whether the C-reactive protein (CRP) is a trustworthy marker in detecting inflammation in children and adolescents with metabolic syndrome.
I read the study with interest. The study is interesting and well designed, although several issues were identified during review. These should be addressed and corrected. My concerns are as follows:
1. Abstract – Confidence interval should be presented as range and p-value as ''p'': E.g. ''95% CI from 0.38 to 1.39, p-value = 0.0006'' should be '' 95% 0.38 - 1.39, p = 0.0006). In remaining text replace ''p-value'' with ''p''.
2. The authors should submit a separate table showing the results of the search strategy, including the number of papers identified in each database, the total number of papers, the number of duplicates, and finally the number of papers after duplicates have been removed.
3. The authors state that the quality assessment of included studies was performed using the Newcastle–Ottawa Quality Assessment Scale (NOS) for observational studies (cross-sectional, case–control, or cohort), scoring them from 1 to 9 but they did not present the results. The authors should provide a separate table showing the results of the assessment of methodological quality and risk of bias.
4. Flow chat of the study is not well designed, it should be composed of four categories: identification, screening, eligibility and included. Because the authors conducted this systematic review in accordance with the PRISMA 2020 statement guidelines, they should use the PRISMA flowchart template available at the following link: http://prisma-statement.org/prismastatement/flowdiagram.aspx. The present flowchart is not appropriate.
5. Paragraph 3.2. χ2(8) = 378.4428 – it is unclear what represents number 8 in brackets. The same follows for paragraph 3.3.
6. Since there are no restrictions on the number of tables and space in MDPI journals, all supplementary tables and figures should be included in the main file.
7. Has the systematic review been registered in the International Prospective Register of Systematic Reviews (PROS-PERO)? This is a standard, and if the study has been registered, the statement including the registration number should be provided.
8. The quality of the English language should be improved. The manuscript should be edited by a native English speaker or a professional language editor to improve grammar and readability.
Comments on the Quality of English LanguageThe quality of the English language should be improved. The manuscript should be edited by a native English speaker or a professional language editor to improve grammar and readability.
Author Response
The authors express their gratitude for the insightful and stimulating comments and suggestions provided. Additionally, we extend our thanks for the careful observation of errors and the effort taken to bring them to our attention, allowing us the opportunity to clarify and rectify the paper accordingly. As a general approach to enhance the clarity and precision of the manuscript, changes were made to its content using the "Track Changes" feature. These revisions were undertaken with a commitment to improving the overall coherence and comprehensibility of the research presented in the paper. We appreciate the valuable input, which has played a pivotal role in refining the quality of our work.
Question 1: Abstract – Confidence interval should be presented as range and p-value as ''p'': E.g. ''95% CI from 0.38 to 1.39, p-value = 0.0006'' should be '' 95% 0.38 - 1.39, p = 0.0006). In remaining text replace ''p-value'' with ''p''.
Answer 1: We sincerely appreciate your feedback and have duly incorporated the suggested changes into the text. To address your recommendation, we have represented the range as an interval between brackets, ensuring accuracy and adherence to the specified format. Thank you for your valuable input, which has contributed to the precision and clarity of our presentation.
Question 2: The authors should submit a separate table showing the results of the search strategy, including the number of papers identified in each database, the total number of papers, the number of duplicates, and finally the number of papers after duplicates have been removed.
Answer 2: These items can be found in Figure 1 (PRISMA flow of the selection process) and we also presented them in lines 215-217. If you would like us to have a separate section in the text addressing this particular issue, please let us know.
Question 3: The authors state that the quality assessment of included studies was performed using the Newcastle–Ottawa Quality Assessment Scale (NOS) for observational studies (cross-sectional, case–control, or cohort), scoring them from 1 to 9 but they did not present the results. The authors should provide a separate table showing the results of the assessment of methodological quality and risk of bias.
Answer 3: Indeed, we made the quality assessment using the Newcastle–Ottawa Quality Assessment Scale and we displayed the results in the last column of Table 1. Unfortunately, we forgot to mention it in the text, so we are very grateful for your kind observation. We made this change in the text: “The results are displayed in Table 2.” The number of the table changed due to the fact that we added another table in the text.
Question 4: Flow chat of the study is not well designed, it should be composed of four categories: identification, screening, eligibility and included. Because the authors conducted this systematic review in accordance with the PRISMA 2020 statement guidelines, they should use the PRISMA flowchart template available at the following link: http://prisma-statement.org/prismastatement/flowdiagram.aspx. The present flowchart is not appropriate.
Answer 4: In our manuscript, we followed the PRISMA Statement disseminated in BMJ as stated in the reference [29] in our manuscript. According to this new statement, the new PRISMA flowchart includes only three categories: Identification, Screening and Included.
Page, M.J.; McKenzie, J.E.; Bossuyt, P.M.; Boutron, I.; Hoffmann, T.C.; Mulrow, C.D.; Shamseer, L.; Tetzlaff, J.M.; Akl, E.A.; Brennan, S.E.; et al. The PRISMA 2020 Statement: An Updated Guideline for Reporting Systematic Reviews. BMJ 2021, n71, doi:10.1136/bmj.n71.
Question 5: Paragraph 3.2. χ2(8) = 378.4428 – it is unclear what represents number 8 in brackets. The same follows for paragraph 3.3.
Answer 5: The requirements for presenting chi-square test results include the degrees of freedom, which is the number of studies minus 1. We have included 9 studies in the meta-analysis from paragraph 3.2, so the degrees of freedom are 8. We have included 12 studies in the meta-analysis from paragraph 3.3, so the degrees of freedom are 11, and the same for the other paragraphs.
Question 6: Since there are no restrictions on the number of tables and space in MDPI journals, all supplementary tables and figures should be included in the main file.
Answer 6: Thank you for your suggestion. We have a lot of figures and tables to add to our study. We did not add all of them in order to not be too overwhelming for the reader. To your suggestion, we will add them to the main text.
Question 7: Has the systematic review been registered in the International Prospective Register of Systematic Reviews (PROS-PERO)? This is a standard, and if the study has been registered, the statement including the registration number should be provided.
Answer 7: Thank you for addressing this issue. Unfortunately, at the beginning of our study, we were not aware of this requirement and did not register our manuscript in the PROSPERO database. Anyhow, we have verified the PROSPERO database and no systematic review like ours has been registered.
In order to address this issue, the editors suggested us to register our study in the INPLASY platform, which we have already done. Our registration number is INPLASY2023100032 and our DOI is 10.37766/inplasy2023.10.0032.
We also added this information in the text:
“In our research, we followed the 27 items for the Systematic reviews and Meta-Analysis (PRISMA) and the flow chart was designed according to these guidelines [29]. We registered our work using the INPLASY platform which is an international platform of registered systematic review and meta-analysis protocols. Our identification number is INPLASY2023100032 and our DOI is 10.37766/inplasy2023.10.0032.”
Question 8: The quality of the English language should be improved. The manuscript should be edited by a native English speaker or a professional language editor to improve grammar and readability.
Answer 8: Thank you for pointing this out! We made some adjustments in the text in order to address this issue and also requested a native speaker to proofread our English.
Round 2
Reviewer 1 Report
Comments and Suggestions for Authors
I think the authors submitted meticulous responses and corrections.
Thank you for getting it resolved my requirement. I don't have any additional comments or requests.
Thank you.
Reviewer 3 Report
Comments and Suggestions for Authors
The authors responded appropriately to my comments and improved the manuscript. I have no further comments or requests.
Comments on the Quality of English LanguageModerate editing of English language required